# Diagnostic Accuracy of the Overlapping Infinity Loops, Wire Cube, and Clock Drawing Tests in Subjective Cognitive Decline, Mild Cognitive Impairment and Dementia

**DOI:** 10.3390/geriatrics7040072

**Published:** 2022-07-05

**Authors:** Sigourney Costa, Rebecca J. St George, James Scott McDonald, Xinyi Wang, Jane Alty

**Affiliations:** 1School of Medicine, University of Tasmania, Hobart 7001, Australia; sccosta@utas.edu.au; 2Wicking Dementia Research and Education Centre, University of Tasmania, Hobart 7001, Australia; rebecca.stgeorge@utas.edu.au (R.J.S.G.); js.mcdonald@utas.edu.au (J.S.M.); xinyi.wang@utas.edu.au (X.W.); 3School of Psychological Sciences, University of Tasmania, Hobart 7005, Australia; 4Royal Hobart Hospital, Hobart 7001, Australia

**Keywords:** dementia, mild cognitive impairment, subjective cognitive decline, figure drawing task, Addenbrooke’s Cognitive Examination, clock drawing test, wire cube, infinity loops

## Abstract

Figure drawing tasks are commonly used standalone or as part of broader screening tests to detect cognitive impairment. Only one study has compared the classification accuracy of three common drawing tasks—overlapping infinity loops, wire cube, and the clock drawing task (CDT)—in mild cognitive impairment (MCI) and dementia, but age and education, which impact performance, were not accounted for. We replicated the research, adjusting for age and education and, for the first time, assessed subjective cognitive decline (SCD) too. Participants were recruited from the Tasmanian ISLAND Cognitive Clinic and healthy controls from a community sample. All participants completed the three figure drawing tasks. The clinic patients were categorised according to interdisciplinary consensus diagnosis. Binomial logistic regression and area under ROC curves (AUC) were calculated to determine the discriminatory ability of each drawing task. Overall, 112 adults were recruited; 51 had normal cognition (NC), 21 SCD, 24 MCI, and 16 had dementia. The infinity loops test did not discriminate any of the groups, casting some doubt on its usefulness. The wire cube discriminated NC from dementia (AUC 0.7; *p* < 0.05). The CDT discriminated NC from dementia (AUC 0.77; *p* < 0.01), NC from cognitive impairment (dementia + MCI; AUC 0.59; *p* < 0.05), and MCI from dementia (AUC 0.76; *p* < 0.01). None of the tests discriminated NC from MCI or NC from SCD. The CDT was the most discriminatory test, followed by the wire cube. This may help guide clinicians who often choose just one figure drawing task due to time constraints or patient fatigue.

## 1. Introduction

As people live longer, the incidence of mild cognitive impairment (MCI) and dementia is increasing. There is an urgent need to develop brief screening tests that can accurately detect cognitive impairment and early-stage dementia [1]. This would facilitate referral for further assessment, timely diagnosis and early interventions, such as risk factor modification and recruitment to drug trials.

Figure drawing tasks require visuoconstructional abilities, which incorporate different cognitive abilities, including visuospatial processing, executive function and planning, and fine motor skills [2,3,4]. The temporal and parietal lobes of the brain are considered central to these abilities. In Alzheimer’s disease, which is the most common neurodegenerative dementia subtype, there are early structural changes to the temporal and parietal lobes, resulting in functional impairment [5]. In addition, reduced blood flow to the frontal lobe is associated with a decline in executive function [4,6]. Therefore, the multidomain nature of figure drawing tasks indicates they may be utilised as screening tools for early neurodegeneration. They are commonly used as part of broader cognitive screening tests for dementia. Figure drawing tasks require participants to draw freehand on paper, following verbal instructions or copying an image. There are subtle differences in the diagnostic performance of drawing under instruction compared to image copying in MCI and dementia; drawing under instruction is slightly more discriminatory than image copying, but for the most part, their performance is comparable [7].

There is a range of different figure drawing tests currently used, and it is unclear how they compare in terms of their ability to discriminate MCI and dementia from healthy ageing. This is an important clinical question as clinicians are frequently compelled to use fewer drawing tests in shortened screening protocols to save time and minimise participant fatigue.

The clock drawing test (CDT) is the most commonly used drawing under instruction test and is the most widely researched. It has been suggested that it can detect dementia and potentially MCI to a lesser extent, though there are still inconsistencies in research findings [8]. Additionally, few studies exist that compare figure drawing tests to each other in clinical and prodromal dementia, with only one previous study comparing figure drawing tests in people with MCI [9]. This particular study by Charernboon (2017) determined that the CDT, copying a wire cube and copying overlapping infinity loops could detect dementia, while none detected MCI. There are no extant studies comparing them in people with subjective cognitive decline (SCD). It is recognized that the SCD group is heterogeneous, and not all cases are caused by underlying dementia pathology. However, this group is known to be at higher risk of subsequent MCI and dementia than those without cognitive complaints [10]. Although people with SCD, by definition, have overall normal cognitive test scores in terms of being above a threshold cut-off, it remains unclear if their performance on figure drawing tasks is completely normal; it is possible that they have mildly impaired performance and are intermediate between the healthy controls and MCI group as this set of tasks has never been closely examined before. Therefore, studying the SCD group is an opportunity to understand whether one or more of the figure drawing tasks may be sensitive to the preclinical stage of dementia. It has been suggested that there may be early functional decline in SCD, which causes figure drawing impairment, to the extent that it can predict future MCI [11].

The overlapping infinity loops test is included in Addenbrooke’s Cognitive Examination III (ACE-III) [12], which is one of the most frequently used clinical tests of global cognition. The overlapping infinity loops test was designed to emulate the well-validated interlocking pentagons figure copy test from the Mini-Mental State Examination (no longer open access); however, it had not been validated at the time the ACE-III was proposed [1]. The ACE-III also assesses visuospatial function with the CDT and copying of a wire cube. The ACE-III contains other subtests that assess the cognitive functions that overlap with figure drawing tests. Two additional visuospatial tests include a dot counting test and a letter identification test [12]. Two verbal fluency tests assess executive function, which includes generating words starting with a random letter and an animal naming test [12].

The Charernboon (2017) study suggested that none of the figure drawing tasks could discriminate between normal cognition (NC) and MCI, but all could detect dementia, with CDT having the greatest classification accuracy [9]. These results have not yet been replicated. We know from studies of figure drawing performance in other cognitive disorders, such as Parkinson’s disease with MCI, that the wire cube is more sensitive to MCI than interlocking pentagons [13], but the wire cube and CDT have never been compared outside Charernboon’s study. Furthermore, Charernboon did not adjust for age or years of education, despite the dementia group being older than the MCI and control groups. There is evidence which suggests that increasing age and lower education levels negatively impact figure drawing performance [14]. Notably, the CDT and wire cube are two particular figure drawing tests that have demonstrated the influence of age and education level [2,14,15,16]. This does call into question the veracity of Charernboon’s results; therefore, replicating the study with statistical adjustment for these confounding variables is important.

The objective of this study was to determine the classification accuracy of the overlapping infinity loops, the wire cube and the CDT performance in adults with SCD, MCI, and dementia. We thus aimed to replicate Charernboon’s findings, but with more robust analyses adjusting for age and education, and also to extend the work to include an SCD group. This is the first study to test the ability of the figure drawing tests to differentiate between NC and SCD.

## 2. Materials and Methods

### 2.1. Participant Recruitment

Participants were recruited from adults attending the ISLAND Cognitive Clinic at the University of Tasmania, Australia, between April and December 2021. The clinic provides a “one-stop” interdisciplinary assessment and consensus diagnostic service to any adults who reside in Tasmania. The inclusion criteria were that all participants were eighteen years or older and had experienced persistent cognitive symptoms for more than 3 months without a reversible cause. The participants were excluded if they had acute behavioural disturbances, acute psychosis or other acute psychiatric disturbance, significant ongoing substance abuse or were at a crisis point from the carer’s perspective. All participants provided video and written informed consent for their clinical and demographic data to be analysed for research purposes. Ethics approval was provided by the University of Tasmania Human Ethics Committee (reference 18639).

### 2.2. Clinical Diagnosis

Participants attending the ISLAND Cognitive Clinic were assessed by a board-certified medical specialist (neurologist or geriatrician) and had detailed neurocognitive assessments undertaken by a board-certified psychologist. The ACE-III was administered by the medical specialist as part of their standard clinical assessment. The participants underwent an array of ancillary tests, including an MRI brain scan and a range of blood pathology tests. The assessments from each participant were discussed at a multidisciplinary conference team meeting, and a consensus diagnosis was formed in line with gold-standard guidelines and criteria [10,17]. The clinic participants were classified as having dementia, MCI or SCD [18].

### 2.3. Control Group

The control group consisted of participants who were recruited from a longitudinal community study of healthy ageing and dementia prevention in Tasmania, The ISLAND Project [19]. The inclusion criteria were that all participants were eighteen years or older and did not have any persistent cognitive symptoms over the previous year. Anyone with a diagnosis of cognitive impairment or dementia was excluded. The ACE-III was administered to healthy controls by a neurologist, psychologist or trained researcher as part of their standard research assessments. All assessments for both the clinic participants and the community study participants were held in the clinic rooms at the Clinical Research Facility of the University of Tasmania in the Medical Sciences Precinct, Hobart.

### 2.4. Figure Drawing Tests

The figure drawings were scored according to the standard ACE-III criteria, as follows. The overlapping infinity loops drawings were scored as either 0 or 1, with 0 points being incorrect and 1 point being correct. The wire cube had a score range of 0 to 2, with 0 points for an incorrect cube, 1 point for a general cube shape with fewer than 12 lines, and 2 points for a correct cube. The CDT had a score range of 0 to 5; with 1 point for a circle, 1 point for all the numbers, 1 point if all numbers are appropriately distributed within the circle, and 1 point each per correctly placed hand representing the time ten past five (see Figure 1).

### 2.5. Statistical Analysis

One-way ANOVAs with Tukey HSD post-hoc comparisons were used to compare age and years of education between groups. Chi-squared tests were used to compare gender proportions and the other ACE-III tests associated with executive function (verbal fluency of words beginning with a particular letter and animals) and visuospatial function (dot counting and letter identification sections) between groups. Due to the negatively skewed ACE-III distribution, a non-parametric Kruskal–Wallis test with Bonferroni-corrected pairwise comparisons determined differences in ACE-III scores between groups. Nonparametric Spearman correlations analysed the association between figure drawing performance and age and years of education across all subjects.

Binomial logistic regression models investigated how well each drawing task predicted cognitive group as follows: NC from SCD, NC from MCI, NC from dementia, NC from those with overarching cognitive impairment (MCI and dementia), MCI from SCD, and MCI from dementia. Age and years of education were included as covariates. The area under Receiver Operating Characteristic (ROC) curves, AUC, were used to determine the optimal sensitivity and specificity of each test for distinguishing groups. SPSS (Version 27.0. Armonk, New York, NY, USA) was used for statistical analysis, and *p* values of less than 0.05 were considered to indicate statistical significance.

## 3. Results

### 3.1. Participant Demographics and Clinical Diagnosis

There were 112 participants included in the study (mean age 68.71 years (SD 9.18), 30.36% male); see Table 1. The number of participants in each classification category was as follows: 51 NC, 21 SCD, 24 MCI, and 16 for dementia. The dementia group comprised nine people with AD (56.25%), six people with mixed (AD and vascular) dementia, and one with vascular dementia (VaD).

The groups of participants based on cognitive diagnosis differed in terms of age, years of education, and cognitive test scores, as outlined in Table 1. The dementia group were significantly older than the other groups (*p* < 0.01 in each case). The NC group had significantly more years of education than the MCI (*p* < 0.001) and dementia groups (*p* < 0.001). The gender proportions did not differ between groups (Chi-Squared = 2.06, *p* = 0.56). As expected, the dementia group had significantly lower ACE-III scores compared to all other groups (*p* < 0.001 for NC and SCD groups; *p* = 0.042 for the MCI group). The MCI group also had significantly lower ACE-III scores compared to both NC and SCD groups (*p* < 0.001 in each case). The verbal fluency tests (letters and animal naming) were significantly worse in the dementia group compared to all other groups.

The number of years of education was associated with better performance on all figure drawing tasks: the CDT (Spearmans’s rho, *p* = 0.20, *p* = 0.035), the infinity loops (*p* = 0.21, *p* = 0.031), and the wire cube drawing (*p* = 0.19, *p* = 0.045) tests. Older participants tended to show worse performance on figure drawing although this did not reach a statistical difference for any of the tasks: CDT (*p* = −0.14, *p* = 0.17), infinity loops (*p* = −0.13, *p* = 0.19), and wire cube (*p* = −0.04, *p* = 0.67). 

### 3.2. Figure Drawing Task Discriminatory Capacity

Both the CDT and wire cube drawing tests could distinguish between the NC and dementia groups when controlling for age and years of education. The CDT test had an odds ratio (OR) of 11.35 and 95% confidence interval (95%CI) = [1.73:74.55], *p* = 0.011, which suggests every point lost on the CDT test was associated with 11.3 times the chance of having dementia. Every point lost on the wire cube test was associated with 7.9 times the chance of having dementia (OR = 7.91, 95%CI [1.31:47.8], *p* = 0.024). The CDT was also able to distinguish between MCI and Dementia groups (OR = 5.04, 95%CI [1.63:15.80], *p* = 0.005) and between NC and the cognitive impairment (MCI and dementia) group (OR = 2.10, 95%CI [1.01: 4.26], *p* = 0.046). No other tests or comparisons were significant. ROC curves of the probability outputs of the binomial logistic regression models (accounting for age and education) are presented in Figure 2, and the raw drawing test scores are presented in Figure 3.

Area under the ROC curve (AUC) calculations confirmed that the CDT was the most discriminatory between cognitively healthy controls and participants with dementia; a score of four or below showed 79% sensitivity and 69% specificity. The CDT also discriminated between MCI and dementia patients, with a score of four or below showing 71% sensitivity and 69% specificity, or with a score of three or below showing 98% sensitivity and 56% specificity. The wire cube also discriminated between NC and dementia, with a score of one or below showing 88% sensitivity and 50% specificity. None of the three tasks discriminated between NC and MCI or between NC and SCD.

## 4. Discussion

The objective of this study was to determine the classification accuracy of the overlapping infinity loops, the wire cube, and the CDT in adults with SCD, MCI, and dementia. We found that the CDT was most discriminatory between healthy participants and those with dementia. Both the wire cube and the CDT could discriminate MCI from dementia. None of the tasks discriminated MCI, or SCD, from healthy participants. The overlapping infinity loops did not discriminate between any of the groups.

These findings partially align with those from the Charernboon (2017) study, the only other study that has examined the figure drawing tests across a range of cognitive groups: healthy controls, MCI, and dementia. Similar to our study, Charernboon determined that none of the tests could discriminate MCI from NC and that the CDT and wire cube could discriminate dementia from NC with a sensitivity/specificity of 76.6%/87.4% (our study, 79%/69%) and 93.6%/46.3% (our study, 88%/50%), respectively. However, Charernboon also found that the infinity loops could discriminate dementia from NC with a sensitivity/specificity of 63.8%/91.6% [9], whereas we determined that the infinity loops task did not discriminate between any of the cognitive groups.

Our study demonstrated that age and years of education alone distinguished between cognitive groups (Figure 2). There is an age-related decline in healthy adults in the cognitive abilities involved in visuoconstruction, including visuospatial ability, executive function, and fine motor skills [20,21,22]. In particular, increasing age is known to have a negative effect on CDT and wire cube performance [2,14,15,16]. Though the effect of age on the infinity loops has not been specifically investigated, the fact that there is a decline in visuoconstructional ability is a sufficient motivator to adjust for this variable in the statistical analyses for all figure drawing tasks. There is also evidence that education impacts figure drawing performance, with several studies demonstrating the effect on CDT and wire cube performance [14,15,16,23]. The cognitive reserve hypothesis suggests that higher education levels have a protective effect on age-related cognitive decline and that those with lower education levels have a reduced ability to compensate for cognitive dysfunction [23,24]. In addition, the acquisition of higher education is associated with better performance across many cognitive domains, including those involved in figure drawing performance such as visuospatial function and fine motor movements [23]. Similar to age, the effect of education on infinity loop performance has not been specifically investigated, but there is concern about its impact on the CDT [25], which encourages the use of adjusted analyses for all figure drawing tests.

In line with prior literature, our results show that the number of years of education was associated with better figure drawing performance across all three figure drawing tasks. Older people tended to have lower scores, although there was not a statistical difference in our sample. To remove these potentially confounding effects, we accounted for years of education and age in the statistical analysis. These adjustments may account for the differences in our findings and Charernboon’s, which did not account for these factors (despite the age differences between clinical groups).

Our results call into question the usefulness of the overlapping infinity loops test. This test was added into ACE-III to replace the well-validated interlocking pentagons but has not previously been validated itself. In our sample, the vast majority of each group could complete this task correctly. This suggests the task is too easy to challenge visuoconstructional ability, even for those with substantial cognitive impairment.

These results indicate that the optimal figure drawing test selection is context-dependent. The CDT would be most appropriate if dementia were suspected, followed by the wire cube. The CDT could also be used to discriminate MCI from dementia; the ability for figure drawing tests to detect this has not been explored in prior literature. Lastly, discriminating MCI from NC would be useful, and it is surprising that none were able to, given the CDT and wire cube can discriminate dementia from NC (which is a similar relationship to MCI versus NC). However, it should be emphasised that figure drawing tests are rarely utilised as independent screening tools for dementia, forming the visuospatial or visuoconstructional assessment component of different cognitive screening tests such as the ACE-III, Mini-Mental State Examination, and Montreal Cognitive Assessment.

A recent meta-analysis by Bat et al. [7] investigated the diagnostic accuracy of drawing under instruction tests (e.g., tests such as CDT) compared to image copying tests (e.g., tests such as wire cube and infinity loops) in MCI and dementia across 92 studies. They determined that the two styles of drawing tests were comparable in their ability to detect dementia. They also found that both styles of test detected MCI with higher AUCs for the drawing under instruction tests. This suggests that the CDT, as a drawing under instruction task, has an advantage over the other tests, which supports our findings. However, given Bat et al.’s findings, it is surprising that none of the figure drawing tasks in this study detected MCI. Perhaps this was because different figure drawing tests were pooled in the meta-analysis, with the wire cube and infinity loops forming only a low percentage of the image copying tests (6% and 5% of the total, respectively), which makes it difficult to determine the discriminatory ability of individual tests. A large percentage of the studies included were within the region of Asia, while a few were Australian studies; therefore, cultural differences may account for some of the differences seen. It is also unclear what scoring systems were used across the 92 studies. Lastly, Bat et al. commented that there was potential for bias in patient selection for some studies included, which may impact their pooled results.

This is only the second time that the overlapping infinity loops and wire cube have been examined in MCI, but the CDT has been investigated previously. One study by Ehreke et al. suggests that the CDT is not a good screening instrument for MCI as it is currently administered, yielding a sensitivity/specificity of 76%/58%. They suggest that increasing the complexity of the scoring system may improve its accuracy [26]. A systematic review evaluating nine studies (involving people with a diagnosis of MCI where the CDT is utilised as a screening tool) also suggests the CDT cannot reliably detect MCI [27]. In contrast, Amodeo et al., who examined fourteen studies, including one which evaluated 40 MCI participants and another with 2004 NC participants, found that the CDT is sensitive to cognitive decline over time, with the ability to differentiate at baseline between the NC group who will develop dementia two years post-baseline, as well as between patients with MCI who will progress to dementia up to six years post-baseline [28]. Likewise, Chan et al. [8] identified 90 studies including NC, MCI, and dementia patients and determined that the CDT can detect dementia and is also fair in detecting MCI. Thus, there are inconsistencies in research findings around the figure drawing tasks, though there are studies supporting our finding that none of the three figure drawing tests are sensitive enough to detect MCI, potentially because the degree of cognitive impairment in this group of participants was very mild, or the test is too easy, or because the visuospatial domain may not be significantly impaired at this stage although other domains are.

This is the first study to investigate whether the figure drawing tests can discriminate between NC and SCD, a condition that, when present in older adults, is associated with a greater risk of developing MCI and dementia [29]. SCD is considered to be a heterogeneous group, but for some, it may represent the earliest transitional clinical stage between normal cognition and MCI, during which the individual experiences a subjective decline in their cognitive capacity, but overall objective cognitive test performance is within normal limits [29]. Our study did not find any detectable difference between NC and SCD groups using the three figure drawing tests, and there are several potential explanations for this; cognitive decline might not be represented in the scored aspects of visuospatial function or only represented in other cognitive domains, or the tests could be insensitive to initial decline from a high baseline. Consistent with this, the SCD group had comparatively more years of education and presumably higher baseline function.

A major strength of this study is that all participants had detailed medical and neuropsychological assessments supported by radiological and pathology investigation, and the diagnosis was formed via an interdisciplinary consensus meeting diagnosis, so the clinical classification categories are robust. The addition of the SCD group, with all participants examined by the same team in the same facilities, also provides additional methodological rigour. Several limitations of the study are also acknowledged: the sample size was relatively small, particularly in the dementia groups, and the dementia group is comprised predominantly of probable AD. Many other dementia types, such as Lewy Body dementia and frontotemporal dementia, were not present in the sample; therefore, the figure drawing tests’ ability to detect these other forms of dementia cannot be evaluated. Finally, this study is cross-sectional rather than longitudinal; it would be useful to repeat these tests, especially in the MCI and SCD groups, to establish whether a decline in the assessments is more predictive than a single score.

An avenue of future research would be to compare the performance of these brief figure drawing tests to more complex and well-validated figure copy tests such as the Rey–Osterrieth Complex Figure test, which may have a greater ability to discriminate MCI from NC. Examination of the association between figure drawing tasks to MRI and fluid biomarkers of dementia may also be considered for future research.

## 5. Conclusions

This study demonstrates that the CDT is a useful and brief screening tool for detecting a decline in visuoconstructional ability associated with neurodegeneration, as it could independently discriminate NC from dementia and NC from cognitive impairment (MCI and Dementia). Both the CDT and wire cube are useful brief screening tools for discriminating MCI from dementia, but none of the drawing tests could discriminate between NC and SCD, nor NC and MCI. The results question the usefulness of the overlapping infinity loops, which did not demonstrate any discriminatory capacity between the classification categories. Overall, these results indicate that the utility of these figure drawing tests depends on the clinical context. However, these figure drawing tests are rarely used as independent screening tools; therefore, they should still be considered in the context of the rest of the cognitive screening test results.

## Figures and Tables

**Figure 1 geriatrics-07-00072-f001:**
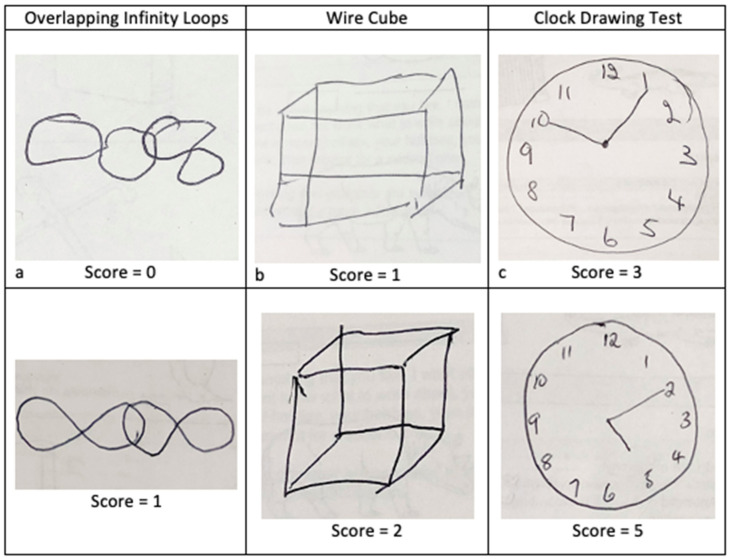
Example drawings of the overlapping infinity loops, wire cube and clock drawing task. (**a**) shows an incorrect drawing with one point lost for incorrect orientation; (**b**) shows a general cube shape is maintained but one point lost for fewer than 12 lines. (**c**) shows a correct clock face (circle and distribution of numbers), but two points are lost for incorrect placement of both hands.

**Figure 2 geriatrics-07-00072-f002:**
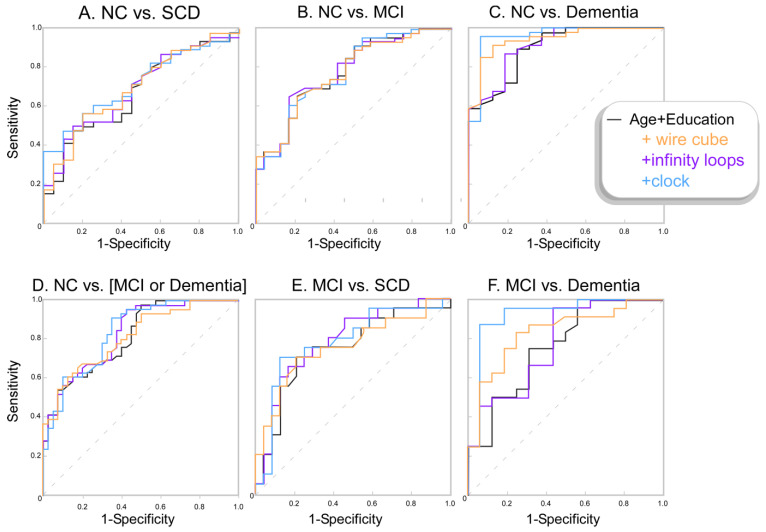
ROC curves showing the sensitivity and specificity of the different binomial logistic model probabilities. The model including only age and years of education is shown in black. The model with the wire cube test included (i.e., age and education and wire cube score) is shown in orange. The model with infinity loops included (age and education and infinity loops score) is shown in purple, and the model with the CDT (age & education & CDT score) is shown in blue.

**Figure 3 geriatrics-07-00072-f003:**
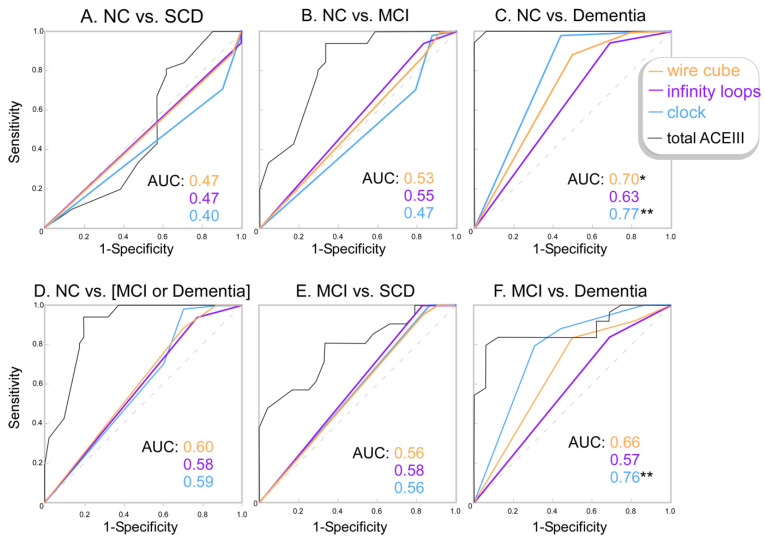
ROC Curves and the corresponding area under the curve (AUC) for drawing tests scores. * *p* < 0.05; ** *p* < 0.01 if the AUC was significantly greater than the diagonal line (0.5) the test did distinguish between the two groups.

**Table 1 geriatrics-07-00072-t001:** Clinical and demographic characteristics of each cognitive diagnosis group. NC is normal controls; SCD, subjective cognitive decline; MCI, mild cognitive impairment; ACE-III, Addenbrookes Cognitive Examination- III; CDT, clock drawing task; letters, verbal fluency test 1; animals, verbal fluency test 2; dot counting, visuospatial test 4; letter identification, visuospatial test 5. *p*-values represent the main effect of group and between-group differences as shown with the following symbols: # *p* < 0.05 significant difference in comparison with all other groups; * *p* < 0.05 significant difference in comparison with NC.

	NC	SCD	MCI	Dementia	*p*-Value
*n*	51	21	24	16	
Age (years): mean (SD)	67.47 (7.63)	65.48 (9.38)	68.50 (10.24)	77.25 (7.38) #	<0.01
Male (%)	25.49	28.57	41.67	31.25	0.56
Total years of education: mean (SD)	16.24 (3.80)	15.05 (3.73)	12.33 (3.16) *	10.56 (2.56) *	<0.01
Total ACE-III score: mean (SD) [range]	95.98 (2.87) [89–100]	95.05 (5.06) [84–100]	88.00 (8.72) # [69–98]	73.38 (9.00) # [56–89]	<0.01
Letters score: mean (SD) [range]	5.98 (1.29) [2–7]	5.95 (1.43) [1–7]	5.46 (1.69) [0–7]	3.73 (2.02) # [0–7]	0.01
Animals score: mean (SD) [range]	6.18 (0.95) [4–7]	6.24 (1.14) [2–7]	5.08 (2.06) [0–7]	3.80 (1.61) # [0–6]	<0.01
Dot counting score: mean (SD) [range]	3.94 (0.24) [3,4]	4.00 (0) [0]	3.92 (0.28) [3,4]	3.80 (0.56) [2–4]	0.23
Letter identification score: mean (SD) [range]	4.00 (0) [0]	4.00 (0) [0]	4.00 (0) [0]	4.00 (0) [0]	NA
% correctly completing infinity loops	94.12	100.00	83.33	68.75	
% correctly completing wire cube	88.24	95.24	83.34	50.00	
% correctly completing CDT	70.59	90.48	79.17	31.25	

## Data Availability

Not applicable.

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
