# Peer review of "Diagnostic Accuracy of the Overlapping Infinity Loops, Wire Cube, and Clock Drawing Tests in Subjective Cognitive Decline, Mild Cognitive Impairment and Dementia"

_geriatrics, 2022, doi:10.3390/geriatrics7040072_

Round 1
Reviewer 1 Report
Abstract
1. The authors should present the results in the same order as in the Method (i.e., infinity loops, wire cube, and clock drawing task)
Keywords
2. The keywords do not represent adequately the study (i.e., no mention of MCI and SCD)
Introduction
3. 2nd paragraph: the authors should better explain the motor and cognitive processes involved in drawing tests, and their usefulness in the detection of neurodegenerative diseases.
4. The different cognitive (and motor) processes involved in the three drawing tasks (e.g., visuoconstructional abilities, executive functions) should also be described in order to better appreciate their differential contribution to the diagnosis.
5. The focus on the ACE-III seems inappropriate. The study is not on the clinical validation of this tool, but on the detection of cognitive impairment through the administration of drawing tasks.
6. The authors should avoid the terms “diagnostic accuracy” since SCD is not a disease.
Materials and methods
7. The description of the clinic participants and the control participants should be made using clear inclusion and exclusion criteria.
Results
8. Table 1: add a note to explain the symbols referring to p values.
9. Table 1: considering the cognitive processes involved in drawing tests, consider adding the ACE-III scores on other visuospatial abilities (dots counting; letter identification) and executive functions (verbal fluency) subtests.
10. In the same vein, it would be interesting to perform a correlation analysis between the scores on the three drawing tests and: 1) the total score on the ACE-III; 2) a composite score made of all the visuospatial abilities of the ACE-III; 3) the score on verbal fluency (executive functions).
Discussion
11. There is no need for subtitles in the discussion
12. The authors should refer to the recent meta-analysis conducted by Baker et al. 2022 to discuss their results (Baker K. K. et al. (2022) Comparing drawing under instructions with image copying for mild cognitive impairment (MCI) or dementia screening: a meta-analysis of 92 diagnostic studies, Aging & Mental Health, 26:5, 1019-1026). This article should also be cited in the introduction.
13. The following reference should also be used: Chan JYC et al. (2021). Evaluation of Digital Drawing Tests and Paper-and-Pencil Drawing Tests for the Screening of Mild Cognitive Impairment and Dementia: A Systematic Review and Meta-analysis of Diagnostic Studies. Neuropsychol Rev. doi: 10.1007/s11065-021-09523-2.
14. The discussion should also address the fact that drawing subtests are rarely used alone but are usually part of tests of general cognition such as the MOCA or the MMSE. Their utility for the diagnosis of cognitive impairment associated with pathological aging should be contextualized within these more complete screening tests.
Author Response
Thank you for your expert review and feedback. We address each of your concerns individually within the attached word document.

Reviewer 2 Report
The authors had an original idea to examiner the utility of the infinity loop drawing task of the ACE-III. However, there are several problems with their research design. First, they examined some of the tasks from a domain (visuospatial) that is not typically affected during the early or prodromal stages of Alzheimer's Disease. They analyzed the drawing tasks, but not the the perceptual organization tasks without explaining why. Second, according to some of the references the authors cited, the ACE-III total score is sensitive to detecting MCI from healthy controls, but the drawing tasks are not, and even less sensitive to detecting SCD. Therefore, there is no clear justification for including the SCD group in the analyses, especially because by definition, people with SCD don't do poorly on objective cognitive tests. Similarly, there's no clear rationale provided for examining scores on the drawing tasks separately from the ACE-III total score.
An investigation into the utility of the infinity loops task is a worthwhile one based upon the background information the authors provided, as well as their ROC analyses examining the value added to AUC calculations by including that task. The significance of their research would be greater if they examined the association between performance on the drawing tasks and MRI and fluid biomarkers for AD, rather than focusing on the tasks' ability to detect SCD or MCI. Their argument that the infinity loops task is not a good indicator of visuospatial function would also be stronger if it was compared to participants' performances on other tests given in their clinic.
Author Response

(The authors gave the same response as above.)

Reviewer 3 Report
Thanks for recommending me as a reviewer. In this paper, authors replicated the research, adjusting for age and education and, for the first time, assess subjective cognitive decline (SCD). Participants were categorised according to interdisciplinary consensus diagnosis. In this study, binomial logistic regression and area under ROC curves (AUC) were calculated to determine the discriminatory ability of each drawing task. If authors complete minor revisions, the quality of the study will be further improved.
1. The introduction section is well written. If the authors describe the trends of prior research in more detail in the introduction section, it can help readers understand.
2. The quality of the study would be further improved if the authors separately indicated the legend in Table 2.
3. The quality of the study would be improved if the authors were more specific in the conclusion section.
Author Response
Thank you very much for your expert review, positive comments and helpful suggestions. We address each of your concerns individually within the attached word document.

Round 2
Reviewer 1 Report
The authors have very well responded to all my comments and suggestions. I congratulate them for that and for the quality of their study.
Minor comment.
Table 1. symbols for statistical differences in the scores on the CDT and the wire cube subtests are missing in Table 1.
Author Response
Thank you for your expert review, feedback and positive comments. Below we address your remaining concern.
The authors have very well responded to all my comments and suggestions. I congratulate them for that and for the quality of their study.
Minor comment.
Comment 1: Symbols for statistical differences in the scores on the CDT and the wire cube subtests are missing in Table 1.
Author reply: The appropriate symbols indicating statistical significance have now been inserted into Table 1.
Reviewer 2 Report
Table 1 is missing p-values and superscripts for all the new variables and comparisons added to the revised version. As a result, I cannot comment on the pairwise differences in subtest scores, which may be meaningful and worth highlighting in the Discussion.
The authors cite the lack of adjustment for age and education as a weakness of the Carernboon (2017) paper, but they didn't spend any time discussing how either of those variable would impact visuomotor performance or diagnostic categories in the Introduction nor Discussion section. The authors may want to run their analyses without adjusting for those two variables to see if they find more group differences with the increased degrees of freedom, and provide results from both models (with and without covariates). As it stands, their findings support Carernboon's and doesn't convince the reader of a need to control for age and years of education.
The sentence in lines 48-50 is worded awkwardly. I suggest dividing it into two sentences.
Lines 239-241 is confusing, it sounds like the two tests can do the same thing.
Lines 243-246 seem to contradict the purpose of their study. If figure drawing tests are not used independently, then why examine them independent of the total score?
The paragraph describing Bat et al.'s meta-analysis should provide further explanation for why the current study's results were inconsistent with the 92 studies reviewed. What is unique about this study's sample? Is it culturally different than the other samples studied? Is it a methodological difference because this study used a more rigorous approach to diagnostic classification or used different operational definitions for MCI?
Lines 314-315 are a repeat of lines 239-241.
Author Response
Thank you for your expert review and feedback. We address each of your remaining concerns within the attached document.
